# The Application of the Ultrasound Technique in the Production of Rosé and Red Wines

**Victoria Lizama \*** 📷, **Inmaculada Álvarez and María José García-Esparza**

Instituto de Ingeniería de Alimentos-FoodUPV, Universitat Politècnica de València, Cmno de Vera s.n, 46022 Valencia, Spain; inmalva@tal.upv.es (I.Á.); mesparza@tal.upv.es (M.J.G.-E.)
\* Correspondence: vlizama@tal.upv.es

**Abstract:** The application of the ultrasound technique (US) in the production of rosé and red wines has demonstrated that the aromatic composition of rose wine can be affected and that it contributes to increasing the color of red wines without increasing the extraction of astringent tannins. The ultrasound treatment has favored the extraction of anthocyanins, which has had an impact on the increase in color density (C.D.) and has allowed greater color stability over time. Moreover, significant differences have been found between the two US systems applied, with continuous treatment being more effective in the extraction of phenolic compounds than pulsed treatment. The application system of the US also affects the aromatic composition of the wines. These results are of interest, as some esters have been described as important odorants in wines.

**Keywords:** ultrasounds; aroma compounds; polyphenolic composition; wine quality





## 1. Introduction

The advance of climate change and growing competition between producers, both on a national and an international level, means that the wine sector is facing the challenge of incorporating new techniques and technological tools that contribute to improving the set of operations involved in the winemaking and conservation processes. In this sense, one of the avant-garde options of considerable scientific interest is the use of ultrasound in winemaking processes. It has recently been approved and recognized by the International Organization of Vine and Wine (OIV), according to Resolution OIV-OENO 616-2019, as a legal practice in relation to enhancing the extractability indices of phenolic compounds obtained from crushed berries [1].

Today's red wine consumers are looking for aromatic products with a lot of color and body that are not too astringent or bitter, remain stable over time and withstand barrel aging well. For this reason, to satisfy these preferences, different techniques have been developed during the maceration process, which aim to increase polyphenolic concentration but with some inconveniences. In this sense, the use of a cold soak prior to fermentation [2] permits reliable results and is very useful in grapes with a lower degree of maturity [3,4]; however, carbonic snow can be costly. On the other hand, the thermovinification techniques (40–70 °C for 15 min to more than 1 h) accelerate extraction, but produce colloidal instability [5]. Furthermore, thermal treatment applied at a low pressure (flash détente) at elevated temperature (85 °C) applied to the grapes for a short time does not affect the pulp but the depressurization in a vacuum cell (40–75 Mbar) causes the boiling of the liquid content in the skin cells, followed by their explosion. This process creates microfissures, facilitating the fast extraction of tannins and pigments, but requires a large economic investment [6].

Of these different extraction techniques, ultrasound is the cheapest, simplest, and most efficient alternative [7]. This technology is based on mechanical waves (inaudible to human ears) with a frequency of >16 kHz. The use of ultrasounds in industrial processes requires two main factors: a liquid medium and a source of high-energy vibrations [8]. The mechanical activity of ultrasound supports the diffusion of solvents into the tissue.

Ultrasound mechanically breaks the wall by shear forces, producing a cavitation effect in the liquid, thus facilitating the transfer of the cell into the solvent. The particle size reduction of ultrasonic cavitation increases the contact surface between the solid and the liquid phase, thus favoring the intensification of the wine flavor through the extraction of phenolic and aromatic compounds [9,10].

Numerous authors have published works regarding wine quality. Cocito et al. [11] achieved high extraction efficiency for aroma compounds in must and wine. For wineries, Bates and Patist [12] developed commercial extraction systems using high-power ultrasound in the food and beverage industry. One particular example in the wine industry uses a 32 kW unit to treat 50 $m^3$/h of must for the extraction of grape color and anthocyanin during the fermentation process. Additionally, oak barrels could benefit from power ultrasound treatment, not only for removing tartrate, but also killing the spoilage microorganisms that are located deep in the pores of the wood. Yap et al. [13] states that this can be an effective decontamination system for *Dekkera/Brettanomyces*.

High-power ultrasound represents an attractive and promising green alternative, complementing $SO_2$ use, in order to reduce or to eliminate spoilage microorganisms present before fermentation or to control and modulate the microbial activity of spoilage or inoculate microorganisms during primary or secondary fermentation [14]. In addition, the ultrasound technique can be applied to wines and cause accelerated aging [15]. It can promote the phenolic compounds polymerization [16]. Ferraretto et al. [8,17] point out that one of the advantages of using ultrasound in winemaking would be the reduction of maceration times by 30 to 43%.

The aim of this work was to evaluate whether cavitation produced by 400 W radiant surface sonication at 25 kHz can cause the degradation of the skin cells, facilitating the extraction of phenolic and aromatic compounds and their precursors. To meet this objective, the grapes will be subjected to the same sonication conditions, varying both the mode of application (continuous or pulsed) and two different treatment times (10 and 20 min). The sonication technique has been applied to obtain red and rosé wines. There are a large number of research works that apply this technique to the production of red wines; however, there is a lack of studies that specifically focus on sonication applied to rosé wines.

## 2. Material and Methods

### 2.1. Microvinifications

Microfermentation assays in triplicate (Figure 1) conducted with Bobal grapes originating from Utiel-Requena D.O. Grapes were harvested manually in boxes (10 kg), destemmed by hand and lightly crushed using a Thermomix blender (model TM 31, Wuppertal, Germany), at high speed for 2 min. The batches were divided into 33 closed glass pots with a 2 kg capacity. Grapes were crushed and subsequently sulfited with potassium bisulphite (E-224, Agrovin, Alcazar de San Juan, Spain) at a rate of 100 mg/kg of grape. Once this process was completed, the assays were subjected to two different treatments: continuous or pulsed sonication via a radiant surface of 400 W to 25 KHz of 10 to 20 min (Figure 1).

The rosé wines came from the samples sonicated for 10 or 20 min, after which they were subjected to the pressing process (placed on a new press) to ferment in the absence of the skins. The control wines were samples without the application of ultrasound and were macerated for 10 or 20 min, respectively, before pressing. To obtain the red wines, the samples with ultrasound treatment were fermented in the presence of the skins for 7 days. The control wine was not subjected to the ultrasound process and was fermented in the presence of the skins for 7 days under the same conditions as the samples subjected to ultrasound. All wines were inoculated with *Saccharomyces cerevisiae* Enartis Ferm Red Fruit (Sepsa-Enartis, La Rioja, Spain).

Once the fermentation process was completed, $SO_2$ was corrected in rosé wines. Each wine was bottled in a glass bottle and 50 mg/L of $SO_2$ was added. On the other hand, lactic bacteria Viniferm Œ104 *Oenococcus oeni* (Agrovin, Alcazar de San Juan, Spain) were inoculated in the red wines. Once the MLF was finished, each wine was bottled in a glass

bottle and 50 mg/L of $SO_2$ was added. All the wines were analyzed 6 months after bottling in order to determine the effect of the treatments conducted on the evolution of the phenolic and aromatic compounds in the wines.

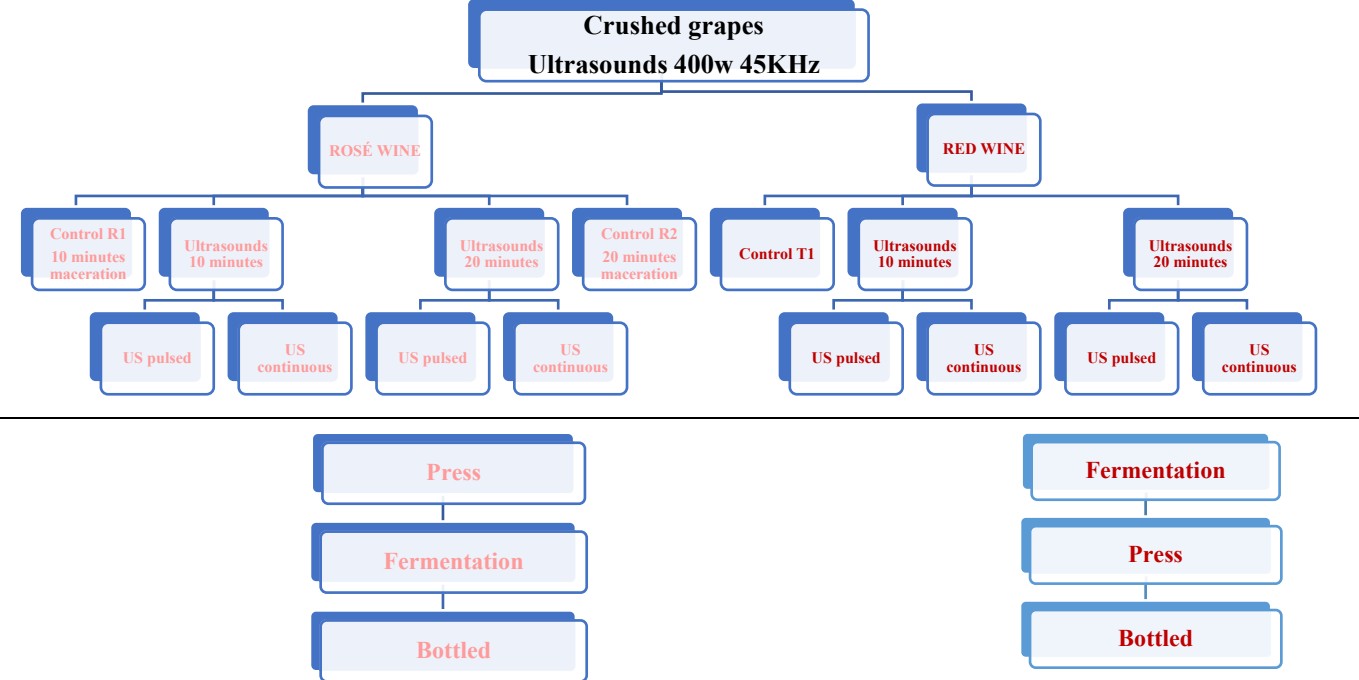

**Figure 1.** Flow diagram.

### 2.2. Analytical Determinations

2.2.1. Physico-Chemical Parameters

The physico-chemical parameters in must and wine (density, ethanol, pH, total and volatile acidity, and $SO_2$ content) were determined according to the official methods established by OIV [18]. Total soluble solids determination was achieved by refractometry and reducing sugars were established by the Fehling method [19].

2.2.2. Phenolic Parameters

The phenolic composition of wine was determined by a JASCO V-630 UV-Visible spectrophotometer (Tokyo, Japan), and a JASCO MD2010 Plus HPLC, coupled with a diode array detector (DAD) (JASCO LC-Net II/ADC, Tokyo, Japan). All the measurements were conducted in triplicate. Color density, Hue, and T.P.I. were estimated by the methods described by Glories [20]. Condensed tannins were determined by the method of Ribéreau-Gayon & Stonestreet [21]. Subsequently, the method reported by Ribéreau-Gayon & Stonestreet [22] was implemented to analyze the anthocyanins. The DMACH index (degree of tannin polymerization) was calculated according to [23].

HPLC was used to quantify individual phenolic compounds (phenolic acids, flavan-3-ols, flavonols, major anthocyanidins, acylated anthocyanins) according to [24]. After centrifugation and filtration, wine samples were injected directly into the HPLC (20 µL). Separation was conducted on a Gemini NX (Phenomenex, Torrance, CA, USA): 5 µm, 250 mm × 4.6 mm i.d. column at 40 °C. The solvents used were 0.1% trifluoroacetic acid (A) and acetonitrile (B). The elution gradient was as follows: 100% A (min 0); 90% A + 10% B (min 5); 85% A + 15% B (min 20); 82% A + 18% B (min 25); 65% A + 35% B (min 30). Individual chromatograms were extracted at 520 nm. For quantification purposes, calibration curves were obtained with malvidin-3-glucoside (S-0911, Extrasynthèse, Genay, France).

### 2.2.3. Volatile Compounds

Volatile compounds were analyzed by the procedure proposed by [25], but with slight modifications. A volume of 2.7 mL of the samples was transferred to a 10 mL screw-capped centrifuge tube containing $4.05 \times g$ of ammonium sulphate (Panreac, Barcelona, Spain) and the following compounds were added: 6.3 mL of milliQ water (Panreac), 20 µL of a standard internal solution (2-butanol, 4-methyl-2-pentanol and 2-octanol from Aldrich, at 140 µg/mL each, in absolute ethanol from LiChrosolv-Merck, Rahway, NJ, USA), and 0.25 mL of dichloromethane (Li-Chrosolv-Merck). The tube underwent mechanical shaking for 120 min and was later centrifuged at $2900 \times g$ for 15 min. The dichloromethane phase was recovered with a 0.5 mL syringe, transferred to the autosampler phial and subsequently analyzed. The chromatographic analysis was conducted in a HP-6890, equipped with a ZB-Wax plus column (60 m × 0.25 mm × 0.25 µm) from Phenomenex. The column temperature, initially set at 40 °C and maintained at this temperature for 5 min, was then raised to 102 °C at a rate of 4 °C/min to 112 °C at a rate of 2 °C/min, to 125 °C at a rate of 3 °C/min and maintained for 5 min and then raised to 160 °C at a rate of 3 °C/min; to 200 °C at a rate of 6 °C/min and then maintained at this temperature for 30 min. The carrier gas was helium, which was fluxed at rate of 3 mL/min. The injection was conducted in the split mode 1:20 (injection volume 2 µL) with a flame-ionization detector (FID detector).

In addition, Kovats retention indices (KI) were calculated for the GC peaks corresponding to the identification of substance by the interpolation of the retention time of normal alkane (C8 eC20) by Fluka Buchs, Buchs, Switzerland), analyzed under the same chromatographic condition. The calculated KI were compared with those reported in the literature for the same stationary phase.

### 2.2.4. Statistical Analysis

In the processing of the results, rosé wines were separated from red wines.

The data for each variable were analyzed with a multifactor analysis of variance (ANOVA), considering the interactions between factors. The effects of the kind of ultrasound treatment (continuous and pulsed) and application time (0, 10, and 20 min) were the factors for this analysis. The statistical significance of each factor under consideration was calculated using the LSD test ($p < 0.05$). The data were statistically analyzed using Statgraphics Centurion XVI (Statpoint Technologies, Inc., Warrenton, VA, USA). The principal component analysis (PCA) was used to examine the grouping of non-treated and treated US wines. PCA and orthogonal projections to latent structures discriminant analysis (OPLS-DA) [26] was performed using SIMCA version 13.0.3 software (MKS Data Analytics Solutions, Malmö, Sweden).

## 3. Results and Discussion

The effect of cavitation produced by radiant surface sonication on the polyphenolic and aromatic composition of rosé and red wines made from the Bobal grape variety following different sonication protocols was studied. This effect on the wines was analyzed 6 months after bottling, comparing the type of ultrasound treatment (continuous or pulsed), as well as different application times (10 and 20 min), considering the control wines (without treatment).

The sugar concentrations of the grapes in the trials ranged between 21.9 and 22.2 °Brix; 5.8–5.9 g/L titratable acidity expressed as tartaric acid; pH 3.7–3.8. The monitoring of the fermentation was conducted on a daily basis by determining the temperature and the density, in order to verify the adequate fermentative kinetics and the absence of stuck fermentation. At a density of 992–993 g/cm$^3$, and when the concentrations of reducing sugars of the wines were between 1.5 and 1.8 g/L, the fermentation process was considered to be complete.

The alcoholic strength of the wines obtained was between 12.8–13.01%, with no significant differences for any of the treatments in these parameters analyzed.

Tables 1 and 2 have been created to compare the treatment effects. They show the mean values and ANOVA of the applied US treatments (without considering the application time) and the control wines (no ultrasound treatment) for rosé and red wines. In addition, Tables 3–6 show the multifactorial analysis of variance (ANOVA) data for the factors "treatment" (continuous, pulsed) and "application time" (10 or 20 min), as well as for the interaction between both, for rosé and red wines. According to the results, the compounds analyzed are affected by the treatment time, especially in rosé wines, and to a lesser extent by the type of ultrasound treatment used (pulsed or continuous).

**Table 1.** Means, standard deviations and variance analyses of the polyphenolic compounds of Bobal rosé and red wines depending on winemaking technology applied. In each row, different letters denote significant differences based on Duncan's test (** $p < 0.05$; *** $p < 0.001$; ns: not significant, n.d.: not detected).

| Compounds | Rosé Wine | | | ANOVA F Ratio | Red Wine | | | ANOVA F Ratio |
|---|---|---|---|---|---|---|---|---|
| | Control Wine | Continuous | Pulsed | Treatment | Control Wine | Continuous | Pulsed | Treatment |
| Color Density | 2.08 ± 0.2 a | 2.99 ± 0.41 c | 2.58 ± 0.3 b | 26.50 *** | 8.77 ± 0.3 a | 10.68 ± 0.3 b | 10.74 ± 0.6 b | 38.56 *** |
| Hue | 88.63 ± 4.6 a | 85.11 ± 3.3 a | 87.94 ± 5.1 a | 2.13 ns | 68.02 ± 2.9 a | 67.6 ± 1.3 a | 68.15 ± 2.5 a | 0.17 ns |
| Total Anthocyanes (mg/L) | 27.91 ± 2.9 a | 44.96 ± 10.1 b | 40.44 ± 7.8 b | 16.47 *** | 368.2 ± 22 a | 392.6 ± 26 ab | 400.46 ± 32 b | 12.58 ** |
| Delphinidin-3-glucoside | 0.28 ± 0.1 a | 0.31 ± 0.04 a | 0.36 ± 0.16 a | 1.68 ns | 14.09 ± 3.9 ab | 15.46 ± 2.5 b | 12.55 ± 1.7 a | 3.79 ** |
| Cyanidin-3-glucoside | 0.16 ± 0.1 a | 0.02 ± 0.01 a | 0.15 ± 0.07 a | 2.36 ns | 2.29 ± 1.5 a | 5.53 ± 0.4 b | 5.31 ± 0.7 b | 34.38 *** |
| Putunidin-3-glucoside | 0.45 ± 0.28 a | 0.50 ± 0.08 a | 0.61 ± 0.1 a | 0.95 ns | 23.80 ± 6.2 a | 26.46 ± 3.1 a | 22.65 ± 3.2 a | 2.58 ns |
| Peonidin-3-glucoside | n.d. | n.d. | n.d. | | 9.91 ± 2.6 a | 12.00 ± 2.6 a | 10.21 ± 1.2 a | 2.95 ns |
| Malvidin-3-glucoside | 4.03 ± 1.6 a | 5.24 ± 2.12 a | 4.02 ± 0.73 a | 2.27 ns | 98.3 ± 20.6 a | 110.3 ± 10.5 a | 102.6 ± 14.0 a | 1.65 ns |
| Condensed Tannins (g/L) | 0.40 ± 0.1 a | 0.41 ± 0.1 a | 0.43 ± 0.1 a | 0.45 ns | 1.14 ± 0.1 a | 1.36 ± 0.1 b | 1.26 ± 0.1 ab | 6.54 *** |
| P.T.I. | 9.61 ± 1.3 a | 11.09 ± 1.5 b | 10.39 ± 0.9 ab | 3.99 ** | 27.9 ± 2.2 a | 42.4 ± 1.1 b | 41.4 ± 3.4 b | 14.70 *** |
| Ethanol Index | 24.9 ± 1.8 a | 29.9 ± 6.9 a | 27.9 ± 4.6 a | 2.11 ns | 33.8 ± 1.0 ab | 36.3 ± 3.9 b | 30.6 ± 2.1 a | 3.49 *** |
| mDP | | | | | 5.44 ± 0.81 a | 5.74 ± 0.37 a | 5.58 ± 0.76 a | 0.46 ns |
| % Galloylation | | | | | 3.57 ± 0.4 a | 4.42 ± 0.4 c | 3.98 ± 0.4 b | 9.16 *** |
| EGC (µM) | | | | | 35.41 ± 9.1 a | 52.33 ± 5.8 c | 44.6 ± 9.0 b | 9.64 *** |
| EPCG (µM) | | | | | 35.7 ± 3.9 a | 43.8 ± 8.7 a | 40.2 ± 3.9 a | 0.01 ns |
| AMW | | | | | 1988 ± 87 a | 2394 ± 141 b | 2317 ± 171 b | 15.89 *** |

**Table 2.** Means, standard deviations and variance analyses of the aromatic compounds of Bobal rosé and red wines depending on winemaking technology applied. In each row, different letters denote significant differences based on Duncan's test ($p < 0.05$).

| Group | Aroma Compound | Rosé Wine | | | Red Wine | | |
|---|---|---|---|---|---|---|---|
| | | Control Wine | Continuous | Pulsed | Control Wine | Continuous | Pulsed |
| Aldehydes | Diacetyl | 156.11 a | 104.7 a | 114.36 a | 204.54 b | 74.22 a | 72.59 a |
| Esters | Ethyl isobutyrate | 521.45 a | 677.71 a | 525.91 a | 596.92 b | 292.04 a | 319.62 a |
| | Isoamyl acetate | 139.78 b | 68.16 a | 63.19 a | 94.61 a | 77.97 a | 126.70 a |
| | Ethyl hexanoate | 399.52 b | 299.46 a | 279.62 a | 462.00 b | 217.00 a | 263.00 a |
| | Hexyl acetate | 21.60 a | 34.87 a | 20.1 a | 15.00 a | 8.00 a | 24.00 a |
| | Ethyl lactate | 1014 b | 545.91 a | 412.04 a | 14.90 b | 7.89 a | 23.76 ab |
| | Ethyl-3- hydroxybutyrate | 382.81 a | 354.82 a | 334 a | 416.00 a | 339.00 a | 336.00 a |
| | Ethyl decanoate | 352.59 b | 313.16 ab | 266.18 a | 346.12 a | 213.79 a | 255.04 a |
| | Diethyl succinate | 382.55 a | 335.29 a | 993.85 a | 569.25 b | 388.53 a | 429.55 ab |
| | Ethyl laurate | 141.19 b | 142.2 b | 89.77 a | 147.44 b | 47.89 a | 104.31 ab |
| **Sum esters** | | 3355.49 | 2771.58 | 2984.66 | 2662.24 | 1592.11 | 1881.98 |
| Alcohols | 1-2 propylene glycol | 76.96 a | 65.39 a | 57.51 a | 537.84 b | 243.67 a | 272.71 ab |
| | Cis-3-hexenol | 667.53 b | 624.15 ab | 425 a | 54.40 b | 28.45 a | 42.82 a |
| | 2-Phenylethanol | 17,802.5 c | 17,376.67 b | 16,995 a | 19,677.17 c | 19,294 b | 18,694.5 a |
| | Isoamyl alcohol | 25,904.63 b | 16,720.61 a | 17,488.1 a | 14,861.97 b | 10,038.98 a | 11,944.21 ab |
| **Sum alcohols** | | 44,451.62 | 34,786.82 | 34,965.61 | 35,131.38 | 29,605.1 | 30,954.24 |
| Terpens | α-pinen | 101.51 a | 202.04 a | 204.36 a | 54.09 ab | 41.35 a | 69.48 b |
| | Linalol | 34.68 a | 68.67 b | 58.66 ab | 66.47 b | 41.76 a | 34.62 a |
| **Sum terpens** | | 136.19 | 270.71 | 263.02 | 120.56 | 83.11 | 104.1 |
| Volatils phenols | 4-vinylphenol | 172.97 a | 156.68 a | 135.16 a | 101.67 b | 62.23 a | 72.95 ab |
| | 2-metoxyphenol (guaiacol) | 769.81 b | 619.18 ab | 403.93 a | 351.38 a | 234.17 a | 286.37 a |
| | 4-ethylguaiacol | 161.27 b | 130.1 ab | 106.77 a | 98.34 a | 66.43 a | 66.87 a |
| | Eugenol | 239.99 a | 255.22 a | 98.06 a | 142.06 b | 106.29 ab | 88.45 a |

**Table 2.** *Cont.*

| Group | Aroma Compound | Rosé Wine | | | Red Wine | | |
|---|---|---|---|---|---|---|---|
| | | Control Wine | Continuous | Pulsed | Control Wine | Continuous | Pulsed |
| **Sum v. phenols** | | 1344.04 | 1161.18 | 743.92 | 693.45 | 469.12 | 514.64 |
| Acids | Decanoic acid | 274.05 a | 253.69 a | 172.76 a | 199.48 a | 165.81 a | 170.95 a |
| | Octanoic acid | 805 a | 935.83 b | 903.33 b | 892.83 a | 1029.42 b | 993.67 b |
| | Hexanoic acid | 469.17 a | 713.83 c | 658.92 b | 498.67 a | 785.22 b | 724.81 c |
| | Butyric acid | 296.08 a | 519.17 c | 458.25 b | 317.53 a | 571.08 b | 504.08 c |
| | Isopentanoic acid | 234.37 a | 352.25 b | 339.67 b | 246.84 a | 387.48 b | 373.63 b |
| **Sum acids** | | 2078.67 | 2774.77 | 2532.93 | 2155.35 | 2939.01 | 2767.14 |
| Others | Theaspirane | 237.39 b | 216.31 b | 141.25 a | 218.44 a | 173.14 a | 150.58 a |
| | Naftalene | 181.48 ab | 145.63 a | 259.79 b | 207.11 a | 256.97 a | 203.35 a |

**Table 3.** Means, standard deviations and variance analyses of the polyphenolic compounds of Bobal rosé wines depending on winemaking technology applied. In each row, different letters denote significant differences based on Duncan's test (** $p < 0.05$; ns: not significant, n.d.: not detected) Tr: treatment, T: time.

| Compounds | Ultrasound Rosé Wines | | | | ANOVA F ratio | | |
|---|---|---|---|---|---|---|---|
| | Continuous | | Pulsed | | Tr | T | Tr*T |
| | 10 min | 20 min | 10 min | 20 min | (Treatment) | (Time) | (Interaction) |
| Color Density | 2.60 ± 0.1 a | 3.38 ± 0.1 b | 2.44 ± 0.1 a | 2.73 ± 0.3 a | 32.97 ** | 57.67 ** | 12.69 ** |
| Hue | 86.87 ± 2.6 a | 83.35 ± 2.6 a | 88.58 ± 1.7 a | 87.30 ± 7.3 a | 2.60 ns | 1.86 ns | 0.41 ns |
| Total Anthocyanes (mg/L) | 38.10 ± 5.1 a | 51.81 ± 9.2 b | 37.0 ± 4.8 a | 43.90 ± 9.0 a | 2.28 ns | 11.92** | 1.29 ns |
| Delphinidin-3-glucoside | 0.32 ± 3.2 a | 0.29 ± 0.0 a | 0.34 ± 0.1 a | 0.39 ± 0.2 a | 1.38 ns | 0.00 ns | 0.73 ns |
| Cyanidin-3-glucoside | 0.04 ± 0.02 a | n.d. a | 0.08 ± 0.05 a | 0.22 ± 0.2 a | 5.89 ** | 0.76 ns | 2.68 ns |
| Putunidin-3-glucoside | 0.53 ± 0.1 a | 0.46 ± 0.0 a | 0.70 ± 0.25 a | 0.53 ± 0.3 a | 0.94 ns | 1.05 ns | 0.15 ns |
| Peonidin-3-glucoside | n.d. | n.d. | n.d. | n.d. | | | |
| Malvidin-3-glucoside | 5.67 ± 1.7 a | 4.82 ± 2.5 a | 4.46 ± 0.8 a | 3.58 ± 0.27 a | 3.54 ns | 1.78 ns | 0.00 ns |
| Condensed Tannins (g/L) | 0.41 ± 0.1 a | 0.42 ± 0.1 a | 0.42 ± 0.1 a | 0.43 ± 0.1 a | 0.24 ns | 0.14 ns | 0.04 ns |
| P.T.I. | 10.12 ± 0.1 a | 12.06 ± 1.1 b | 10.42 ± 0.08 a | 10.34 ± 1.1 a | 12.53 ** | 34.34 ** | 5.43 ** |
| Ethanol Index | 26.96 ± 1.5 a | 29.89 ± 1.6 b | 26.04 ± 1.0 a | 29.74 ± 6.2 a | 3.60 ns | 3.43 ns | 2.52 ns |

**Table 4.** Means, standard deviations and variance analyses of the polyphenolic compounds of Bobal red wines depending on winemaking technology applied. In each row, different letters denote significant differences based on Duncan's test (** $p < 0.05$; *** $p < 0.001$; ns: not significant). Capital letters (A, B) are used to compare treatment influence (continuous, pulsed). Tr: treatment, T: time.

| Compounds | Ultrasound Red Wines | | | | ANOVA F Ratio | | |
|---|---|---|---|---|---|---|---|
| | Continuous | | Pulsed | | Tr | T | Tr*T |
| | 10 min | 20 min | 10 min | 20 min | (Treatment) | (Time) | (Interaction) |
| Color Density | 10.71 ± 0.2 a | 10.65 ± 0.4 a | 10.78 ± 0.6 a | 10.71 ± 0.7 a | 0.07 ns | 0.08 ns | 0.00 ns |
| Hue | 68.81 ± 0.6 a | 66.47 ± 0.5 a | 68.43 ± 2.5 a | 67.87 ± 2.9 a | 0.43 ns | 3.48 ns | 1.30 ns |
| Total Anthocyanes (mg/L) | 396.8 ± 33.1 a | 388.6 ± 20 a | 405.6 ± 29 a | 395.1 ± 38 a | 0.37 ns | 0.56 ns | 0.01 ns |
| Delphinidin-3-glucoside | 16.55 ± 3.2 a | 14.36 ± 0.9 aB | 11.63 ± 1.8 a | 13.7 ± 0.6 bA | 14.27 ** | 0.02 ns | 8.80 ** |
| Cyanidin-3-glucoside | 5.62 ± 0.5 a | 5.44 ± 0.2 a | 5.24 ± 0.9 a | 5.39 ± 0.7 a | 0.75 ns | 0.00 ns | 0.41 ns |
| Putunidin-3-glucoside | 27.85 ± 4.1 a | 25.08 ± 0.9 aB | 20.54 ± 3.3 a | 24.7 ± 1.0 bA | 12.10 ** | 0.44 ns | 10.19 ** |
| Peonidin-3-glucoside | 13.09 ± 3.48 a | 10.90 ± 0.4 aB | 9.05 ± 1.3 a | 11.3 ± 0.7 bA | 5.34 ** | 0.01 ns | 8.53 ** |
| Malvidin-3-glucoside | 114.8 ± 13.5 a | 105.8 ± 3.6 a | 93.8 ± 13.6 a | 111.5 ± 7.6 b | 3.21 ns | 1.05 ns | 9.81 ** |
| Condensed Tannins (g/L) | 1.26 ± 0.1 a | 1.45 ± 0.1 bB | 1.23 ± 0.1 a | 1.28 ± 0.1 aA | 6.43 ** | 9.20 ** | 3.09 ns |
| P.T.I. | 42.1 ± 0.52 a | 42.7 ± 1.4 a | 40.2 ± 4.7 a | 42.61 ± 0.7 a | 0.95 ns | 2.19 ns | 0.80 ns |
| Ethanol Index | 35.4 ± 5.6 a | 37.3 ± 0.7 aB | 29.8 ± 2.8 a | 31.4 ± 0.4 aA | 20.14 *** | 1.88 ns | 0.03 ns |
| mDP | 5.91 ± 0.22 a | 5.56 ± 0.42 a | 5.5 ± 1.1 a | 5.6 ± 0.2 a | 0.41 ns | 0.16 ns | 1.03 ns |
| % Galoylation | 4.30 ± 0.3 a | 4.54 ± 0.5 aB | 3.87 ± 0.6 a | 4.09 ± 0.2 aA | 6.86 ** | 1.86 ns | 0.01 ns |
| EGC (μM) | 52.41 ± 6.1 a | 52.2 ± 6.0 aB | 44.8 ± 13.3 a | 44.4 ± 0.8 aA | 5.87 ** | 0.00 ns | 0.00 ns |
| EPCG (μM) | 39.0 ± 5.0 a | 48.6 ± 9.4 a | 38.2 ± 4.6 a | 42.2 ± 2.0 a | 0.54 ns | 0.68 ns | 0.51 ns |
| AMW | 2379 ± 94 a | 2410 ± 186 a | 2367 ± 217 a | 2268 ± 106 a | 1.39 ns | 0.27 ns | 1.00 ns |

**Table 5.** Means, standard deviations and variance analyses of the aromatic compounds of Bobal rosé wines depending on winemaking technology applied. In each row, different letters denote significant differences based on Duncan's test (** $p < 0.05$; ns: not significant) Tr: treatment, T: time.

| Group | Aroma Compound | Ultrasound Rosé Wines | | | | ANOVA F Ratio | | |
| | | Continuous | | Pulsed | | Tr | T | Tr*T |
| | | 10 min | 20 min | 10 min | 20 min | | | |
|---|---|---|---|---|---|---|---|---|
| Aldehydes | Diacetyl | 134.89 a | 74.52 a | 159.95 a | 68.77 a | 0.06 ns | 3.69 ns | 0.15 ns |
| Esters | Ethyl isobutyrate | 673.11 a | 682.32 a | 489.09 a | 562.73 a | 3.54 ns | 0.26 ns | 0.16 ns |
| | Isoamyl acetate | 68.97 a | 67.34 a | 73.86 b | 52.52 a | 0.56 ns | 2.80 ns | 2.04 ns |
| | Ethyl hexanoate | 318.13 a | 280.79 a | 306.64 a | 252.60 a | 0.33 ns | 172 ns | 0.06 ns |
| | Hexyl acetate | 38.16 a | 31.57 a | 29.68 a | 10.51 a | 2.83 ns | 2.17 ns | 0.51 ns |
| | Ethyl lactate | 531.27 a | 560.54 a | 584.66 a | 239.42 a | 0.85 ns | 1.18 ns | 1.66 ns |
| | Ethyl 3 hydroxybutyrate | 361.22 a | 348.42 a | 365.56 a | 302.43 a | 0.20 ns | 0,69 ns | 0,30 ns |
| | Ethyl decanoate | 330.86 a | 295.46 a | 291.56 a | 240.79 a | 2.13 ns | 1.78 ns | 0.06 ns |
| | Diethyl succinate | 311.29 a | 359.28 a | 320.54 a | 333.28 a | 0.03 ns | 0.41 ns | 0.14 ns |
| | Ethyl laurate | 139.45 a | 144.95 a | 91.53 a | 88.01 a | 4.67 ** | 0.00 ns | 0.04 ns |
| **Sum esters** | | **2772.46** | **2770.67** | **2553.12** | **2082.29** | | | |
| Alcohols | 1-2 propylene glycol | 57.79 a | 72.98 a | 73.40 a | 41.61 a | 0.76 ns | 0.79 ns | 6.49 ** |
| | Cis-3-hexenol | 573.93 a | 674.36 a | 417.06 a | 432.86 a | 3.44 ns | 0.29 ns | 0.16 ns |
| | 2-Phenylethanol | 17,540.00 b | 17,213.33 a | 17,031.67 a | 16,958.33 a | 13.22 ** | 3.63 ns | 1.46 ns |
| | Isoamyl alcohol | 17,557.92 a | 15,883.29 a | 19,630.38 a | 15,345.82 a | 0.16 ns | 2.43 ns | 0.47 ns |
| **Sum alcohols** | | **35,729.64** | **33,843.96** | **37,152.51** | **32,778.62** | | | |
| Terpens | α-pinen | 247.47 a | 156.61 a | 333.81 a | 74.92 a | 0.00 ns | 3.67 ns | 0.85 ns |
| | Linalol | 45.21 a | 92.12 b | 55.15 a | 62.17 a | 0.65 ns | 4.92 ** | 2.74 ns |
| **Sum terpens** | | **292.68** | **248.73** | **388.96** | **137.09** | | | |
| Volatils phenols | 4-vinylphenol | 161.87 a | 151.49 a | 145.19 a | 125.14 a | 1.49 ns | 0.73 ns | 0.07 ns |
| | 2-metoxyphenol (guaiacol) | 640.73 a | 597.62 a | 470.67 a | 337.19 a | 5.30 ** | 0.89 ns | 0.23 ns |
| | 4-ethylguaiacol | 119.81 a | 140.39 a | 138.04 a | 75.50 a | 1.46 ns | 1.13 ns | 4.56 ** |
| | Eugenol | 419.60 b | 90.84 a | 105.91 a | 90.20 a | 9.33 ** | 11.24 ** | 9.25 ** |
| **Sum v. phenols** | | **1342.01** | **980.34** | **859.81** | **628.03** | | | |
| Acids | Decanoic acid | 157.58 a | 349.80 b | 145.41 a | 200.11 a | 4.39 ** | 10.25 ** | 3.17 ns |
| | Octanoic acid | 930.00 a | 941.67 a | 885.00 a | 921.67 a | 3.59 ns | 1.98 ns | 0.53 ns |
| | Hexanoic acid | 739.00 b | 688.67 a | 628.67 a | 689.17 b | 29.72 ** | 0.25 ns | 30.27 ** |
| | Butyric acid | 502.17 a | 536.17 a | 447.17 a | 469.33 a | 22.14 ** | 4.71 ** | 0.21 ns |
| | Isopentanoic acid | 341.33 a | 363.17 a | 323.00 a | 356.33 b | 2.57 ns | 12.36 ** | 0.54 ns |
| **Sum acids** | | **2670.08** | **2879.48** | **2429.25** | **2636.61** | | | |
| Others | Theaspirane | 217.13 a | 215.50 a | 160.98 a | 121.52 a | 6.21 ** | 0.47 ns | 0.39 ns |
| | Naftalene | 223.09 b | 68.17 a | 249.20 a | 270.38 a | 6.58 ** | 2.26 ns | 3.89 ns |

**Table 6.** Means, standard deviations and variance analyses of the aromatic compounds of Bobal red wines depending on winemaking technology applied. In each row, different letters denote significant differences based on Duncan's test (** $p < 0.05$, ns: not significant).

| Group | Aroma Compound | Ultrasound Red Wines | | | | ANOVA F Ratio | | |
| | | Continuous | | Pulsed | | Tr | T | Tr*T |
| | | 10 min | 20 min | 10 min | 20 min | | | |
|---|---|---|---|---|---|---|---|---|
| Aldehydes | Diacetyl | 40.93 a | 107.51 a | 66.18 a | 79 a | 0.00 ns | 2.69 ns | 1.21 ns |
| Esters | Ethyl isobutyrate | 270.68 a | 313.39 a | 167.3 a | 471.94 b | 0.14 ns | 5.63 ** | 3.20 ns |
| | Isoamyl acetate | 50.7 a | 105.23 a | 136.31 a | 117.08 a | 2.71 ns | 0.36 ns | 1.56 ns |
| | Ethyl hexanoate | 216.59 a | 218.3 a | 259.31 a | 266.38 a | 1.36 ns | 0.01 ns | 0.01 ns |
| | Hexyl acetate | 6.6 a | 9.18 a | 33.17 a | 14.35 a | 3.24 ns | 0.84 ns | 1.43 ns |
| | Ethyl lactate | 289.64 a | 121.95 a | 236.86 a | 389.15 a | 1.67 ns | 0.01 ns | 3.72 ns |

**Table 6.** *Cont.*

| Group | Aroma Compound | Ultrasound Red Wines | | | | ANOVA F Ratio | | |
| | | Continuous | | Pulsed | | Tr | T | Tr*T |
| | | 10 min | 20 min | 10 min | 20 min | | | |
| | Ethyl 3 hydroxybutyrate | 350.96 a | 326.59 a | 258.96 a | 412.65 a | 0.00 ns | 1.52 ns | 2.90 ns |
| | Ethyl decanoate | 207.65 a | 219.94 a | 259.17 a | 250.92 a | 1.61 ns | 0.00 ns | 0.10 ns |
| | Diethyl succinate | 358.74 a | 418.32 a | 446.43 a | 412.67 a | 0.43 ns | 0.04 ns | 0.55 ns |
| | Ethyl laurate | 64.78 a | 31.01 a | 126.66 a | 81.96 a | 3.19 ns | 1.54 ns | 0.03 ns |
| **Sum esters** | | **1816.34** | **1763.91** | **1924.17** | **2417.1** | | | |
| Alcohols | 1-2 propylene glycol | 28.26 a | 28.63 a | 32.35 a | 53.29 a | 4.89 ** | 2.77 ns | 2.52 ns |
| | Cis-3-hexenol | 290.17 a | 197.17 a | 221.84 a | 323.58 a | 0.24 ns | 0.01 ns | 2.64 ns |
| | 2-Phenylethanol | 19,294 b | 18,934.67 a | 18,734.83 a | 18,654.17 a | 13.22 ** | 3.63 ns | 1.46 ns |
| | Isoamyl alcohol | 11,877.73 a | 8200.24 a | 10,226.29 a | 13,662.13 a | 1.33 ns | 0.01 ns | 4.62 ** |
| **Sum alcohols** | | **31,490.16** | **27,360.71** | **29,215.31** | **32,693.17** | | | |
| Terpens | α-pinen | 43.94 a | 38.75 a | 79.19 a | 59.77 a | 6.29 ** | 1.21 ns | 0.41 ns |
| | Linalol | 35.69 a | 47.83 a | 41.74 a | 27.5 a | 0.64 ns | 0.01 ns | 2.32 ns |
| **Sum terpens** | | **79.63** | **86.58** | **120.93** | **87.27** | | | |
| Volatils phenols | 4-vinylphenol | 62.32 a | 62.15 a | 72.25 a | 73.64 a | 1.08 ns | 0.00 ns | 0.10 ns |
| | 2-metoxyphenol (guaiacol) | 225.14 a | 243.21 a | 262.46 a | 310.27 a | 1.20 ns | 0.48 ns | 0.10 ns |
| | 4-ethylguaiacol | 78.46 a | 54.39 a | 58.81 a | 74.92 a | 0.00 ns | 0.06 ns | 1.66 ns |
| | Eugenol | 116.71 a | 95.88 a | 80.34 a | 96.55 a | 0.84 ns | 0.01 ns | 0.90 ns |
| **Sum v. phenols** | | **482.63** | **455.63** | **473.86** | **555.38** | | | |
| Acids | Decanoic acid | 149.29 a | 182.32 a | 206.78 a | 135.12 a | 0.02 ns | 0.29 ns | 2.12 ns |
| | Octanoic acid | 1023 a | 1035.83 a | 973.5 a | 1013.83 a | 3.59 ns | 1.98 ns | 0.53 ns |
| | Hexanoic acid | 812.9 b | 757.53 a | 691.53 a | 758.08 b | 29.53 ** | 0.26 ns | 30.10 ** |
| | Butyric acid | 552.38 a | 589.78 a | 491.88 a | 516.27 a | 22.22 ** | 4.68 ** | 0.20 ** |
| | Isopentanoic acid | 375.47 a | 399.48 a | 355.3 a | 391.97 b | 2.57 ns | 12.56 ** | 0.53 ns |
| **Sum acids** | | **2913.04** | **2964.94** | **2718.99** | **2815.27** | | | |
| Others | Theaspirane | 146.22 a | 157.7 a | 200.06 a | 143.46 a | 0.30 ns | 0.38 ns | 0.88 ns |
| | Naftalene | 40.93 a | 204.56 a | 299.31 a | 202.14 a | 2.12 ns | 3.61 ns | 2.41 ns |

On the other hand, a principal component analysis (PCA) was performed using all the chromatic parameters determined, the total concentration of anthocyanins determined by HPLC and, due to the high number of volatile compounds quantified, the sum of the different families of volatile compounds as variables. The aim of this analysis was to find out which variables were responsible for the clustering of the wine samples after the entire winemaking process. This analysis made it possible to reduce the information provided by all the variables studied to two principal components that explain the variability of the data. The results of this analysis showed that the wines were different. The control wines (rosé and red) and those made from sonicated grapes (continuous and pulsed) were separated by PC1, with the control wines having lower values for the chromatic parameters, but a higher content of volatile compounds (esters, volatile phenols, and alcohols), and a lower content of acids. However, it was not possible to separate the wines according to the type of US applied, continuous, or pulsed.

### 3.1. Effect of US Treatment on the Polyphenolic Composition of Bobal Wines

In the rosé wines, according to the findings in Table 1, the ultrasound treatments (regardless of the application time) did not cause significant differences in the polyphenolic compounds analyzed, with the exception of the color density (C.D.), and Total Polyphenol index (T.P.I) and anthocyanin concentration. The ultrasound treatment has favored the extraction of anthocyanins, which has had an impact on the increase in C.D. and has allowed greater color stability over time. All these processes are the results of the increase in tem-

perature that occurs during the cavitation process [16,27]. Moreover, significant differences have been found between the two US systems applied, with continuous treatment being more effective in the extraction of phenolic compounds than pulsed treatment.

On the other hand, in red wines, the application of ultrasound treatment has led to significant differences in all the polyphenolic compounds studied, with lower values found in the control wine than in the wines treated with US prior to fermentation. Wines without US treatment have lower concentrations of anthocyanins, condensed tannins and Total Polyphenol Index; US treatment mechanically breaks the cell envelope and improves the transfer of compounds [27,28] by transforming the structure of the skins, which will yield more compounds during alcoholic fermentation in the presence of the skins. As was the case for rosé wines, US treatment favors the extraction of polyphenolic compounds, in particular by exerting a positive and selective action on the tannic fraction of the grapes, leading to an increase in the percentage of tannin galloylation and epigallocatechin concentration compared to the control wine. Specifically, the percentage of galloylation increases by 24% and 11% with continuous and pulsed US treatment, respectively. The epigallocatechin (EGC) concentration also increased by 47% and 26%, respectively, with continuous and pulsed US compared to the control wine. Epigallocatechin is only found in the skins [29]. In this sense, as the results obtained show, the US technique favors the extraction of tannins from the skins over those from the seeds. These same results were obtained by [30] in a similar trial with the Monastrell variety. Moreover, this result is very interesting, as suggests potential alleviation of the problems that appear when wines are made with grapes with astringent tannins from pips, i.e., with a high percentage of galloyl tannins (EPCG and % galloylation) [31–33], and the ability to reduce maceration times to avoid the extraction of these tannins. However, the mean degree polymerization values (mDP) are not affected.

### 3.2. Effect of US Treatment on the Aromatic Composition of Bobal Wines

Recently, it has been shown that ultrasonic applications on both berries and finished wines could improve aromatic complexity and intensity in wines, either by increasing aromatic compounds or by decreasing aroma-neutralizing substances that provide positive descriptors [11,34–36]. The results of the quantitative analysis of the 27 volatile compounds quantified in the control wine and in the continuous or pulsed sonicated wines are presented in Table 2. The identified compounds include alcohols, esters, terpenes, volatile phenols, and acids.

In rosé wines (Table 2), it was observed that treatment with US applied both continuously and pulsed significantly decreases the concentration of esters such as isoamyl acetate, ethyl hexanoate, ethyl lactate and ethyl decanoate. These results coincide with those obtained by [37], who stated that the concentration of acetates (isoamyl acetate, hexyl acetate) decreases or remains constant in wines made from grapes sonicated at different maceration times. Table 2 also shows that the concentration of some alcohols such as cis 3 hexenol, 2 phenylethanol, and isoamyl alcohol decreases, along with the volatile phenols 2 methoxyphenol and 4 ethylguaiacol. Studies carried out by [38] also show that the application of ultrasound causes a reduction in the concentration of higher alcohols. On the contrary, US treatment has a significant effect on octanoic, hexanoic, butyric, and isopentanoic acids whose concentrations increase in the treated rosé wines, this effect being more noticeable with continuous treatment. Differences were also found between the two US application systems. In rosé wines with continuous application, the concentration of ethyl laurate, 2-phenylethanol, butyric acid, and hexanoic acid significantly increased.

In the case of red wines (Table 2), significant differences were found between the aromatic composition of the control wine and those treated with US: diacetyl, ethyl isobutyrate, ethyl hexanoate, cis 3 hexenol, 2-phenylethanol, and linalool, whose concentrations decreased in the wines treated with US. As with the acids in rosé wines, in red wines treated with US (continuous or pulsed), the concentrations of octanoic, hexanoic butyric and isopentanoic acids increased considerably.

On the other hand, in red wines, the application system of the US also affects the aromatic composition of the wines, since pulsed application caused an increase in the compounds α-pinenen, butyric and hexanoic acids, while continuous application increased the concentration of 2-phenylethanol. These results are of interest, as some esters have been described as important odorants in wines [37]. Short- and medium-chain fatty acid esters, such as ethyl butanoate, ethyl hexanoate, and ethyl octanoate can contribute fruity aromas to wines (strawberry, apple, fruity, and sweet).

As shown in Table 2, the total concentration of esters, alcohols, and volatile phenols in both rosé and red wines is lower in wines treated with US. This can be explained by the fact that the use of high-power and low-frequency ultrasound applied by 400 W radiant surface at 25 kHz caused the spontaneous generation of heat [39], which is one of the side effects of this technology, resulting in the loss of volatiles during the application time.

These results concur with those obtained by [40], who observed that the effect of sonication was significant for the concentrations of esters and acetates, which were similar or slightly lower than those of the control wine. Moreover, these results are in line with those obtained in other studies in which a loss of volatile compounds after ultrasound treatment was observed. As previously discussed, it is believed that this could be due to the loss of volatiles by degassing when increasing the temperature of the ultrasonic system [2,34,41]. However, the latter can be controlled so that the cavitation effect is isolated from the effect of extreme temperature increases [34,35]. However, the results obtained in the present work did not coincide with those obtained by other authors when US treatment was applied during grape maceration. The total ester concentration remained constant or increased, depending on the treatment conditions [35,41,42].

In relation to the total acid concentration (Table 2), it was higher in rosé and red wines sonicated with both continuous and pulsed treatment than in control wines (untreated). Studies by [37] showed that sonication applied to grapes affected fermentation aroma compounds, showing a significant increase in volatile acids. On the other hand, in the case of terpenes, the effect of US is different in the two types of wines produced. The rosé wines subjected to US had a higher concentration of terpenes, as expected; however, the opposite effect occurred in the red wines, with a higher total terpene content observed in the control wines.

### 3.3. Effect of US Application Time on the Polyphenolic Composition of Bobal Wines

According to the results (Tables 3 and 4), the polyphenolic compounds analyzed were more affected by the duration of the treatment time applied (10 or 20 min), especially in the rosé wines, than by the type of US treatment used (pulsed or continuous).

In the case of rosé wines, continuous US treatment affected several of the analyzed compounds, obtaining significantly higher values with a 20 min application than with a 10 min one. The application time caused considerable or significant differences in the compounds C.D., T.P.I., anthocyanin concentration, and ethanol index (related to tannin–polysaccharide bonding). The lowest values of Hue were also obtained following the continuous treatment for 20 min, but without significant differences between any of the treatments, indicating that the level of oxidation is the same for all rosé wines produced 6 months after bottling. Continuous treatment has favored the extraction of anthocyanins, which has an impact on the increase in C.D., which has allowed greater color stability over time. All these processes are due to the increase in Tª produced during the cavitation process [16]; however, other authors [30] obtained an increase in Hue values as a consequence of the increase in Tª caused by US treatment.

In the red wine trial, Table 4 shows the results of the effect of continuous and pulsed US treatment time on the components related to color and astringency. On the one hand, the continuous and pulsed treatments have no significant effect on the variables related to wine color, contrary to what occurred in the rosé wines (Table 3). Nor were significant differences observed in the C.D., total anthocyanins, malvidin-3-glucoside, cyanidin-3-glucoside, mDP, EGC and AMW determinations. The behavior of some detailed anthocyanins should

be highlighted with respect to malvidin-3-glucoside; in this sense, the concentrations of some minority anthocyanins (petunidin-3-glucoside and peonidin-3-glucoside) are affected by the ultrasound treatment applied, achieving higher values in the 20 min ultrasound treatment over the 10 min one.

Depending on the type of US technique applied, the concentration of condensed tannins, ethanol index, percentage of tannin galloylation, and epigallocatechin concentration (EGC) will be affected. In all cases, the values are higher in the continuous treatment than in the pulsed treatment. Continuous treatment causes a higher extraction of compounds because the waves generated do not stop during the application time; however, pulsed treatment is discontinuous, reducing the time of the pauses from the total treatment time, but avoiding an increase in temperature due to the effect of spontaneous heat generation [39]. Continuous treatment is more efficient in the extraction of polyphenolic compounds [8]. A study conducted by other researchers [43] shows that the longer the treatment time, the higher the concentration of polyphenolic compounds and the higher their polymerization; however, in the present study, no significant differences were obtained in the average degree of tannin polymerization. However, as previously mentioned, continuous treatment allows the extraction of more compounds from the skins (determined by the concentration of epigallocatechin) than the extraction of compounds from the seeds, coinciding with the study of [30].

*3.4. Effect of the Time of Application of US Treatment on the Aromatic Composition of Bobal Wines*

The aromatic composition of the wines was determined at the end of the winemaking process, after bottling, in order to conduct a comparative analysis of the results and establish whether the time of application of US gives rise to significant differences in the aromatic compounds of the wines produced. The results of the 27 volatile compounds studied are shown in Tables 5 and 6.

As observed in the tables, the effect of ultrasound treatment time has insignificant effect on the volatile composition of rosé and red wines, especially when continuous treatment is applied.

In relation to rosé wines, the time of continuous US application significantly affects the concentrations of 2-phenylethanol, eugenol, hexanoic acid, and naphthalene, which are higher when applied for 10 min versus 20 min. On the other hand, the concentrations of eugenol and decanoic acid are increased in wines treated with US for 20 min. The application of pulsed US for 20 min produces an increase in hexanoic acid and isopentanoic acid in the wines.

In the trial conducted to obtain red wines, continuous application of US for 20 min caused an increase in 2-phenylethanol and hexanoic acid compared to application for 10 min. When US was applied in pulsed form, there was a significant effect of application time on the concentration of ethyl isobutyrate hexanoic acid and isopentanoic acid, which increased in the wines treated for 20 min.

The time of continuous US treatment affects the volatile composition of rosé wines more than that of red wines. This is due to the fact that red wine production involves maceration during fermentation, which would facilitate the extraction of aromatic compounds during the 7 days of fermentation. In red wines, the effect of US treatment on the aromatic composition has been minimized, while its effect is very evident in rosé wines due to the very short maceration time (minutes).

The time of pulsed US treatment on rosé and red wines has almost no effect on the concentration of volatile compounds. Only hexanoic and isopentanoic acids show higher concentrations when applied for 20 min.

The total concentration of esters, alcohols, terpenes, and volatile phenols was slightly higher in wines treated with US for less time (10 min) with pulsed treatment. However, the total concentration of acids was higher when US was applied for 20 min versus application for 10 min. Studies conducted by [38] showed that when the exposure time to ultrasound treatment increases, the concentration of higher alcohols decreases.

Some critical factors described by [14] in red wines, such as increases in exposure times, affect the concentration of ethyl esters and acetate esters. Lukić et al. [2] point out that elevated temperatures (40 and 60 °C) and prolonged exposure times to US continuously (65 and 90 min) decrease the concentration of esters in general, affecting at the same time the levels of higher alcohols. The cavitation effect could accelerate the degradation rate of higher alcohols in wines [36].

These results are important, since fatty acids and their esters are, together with alcohols, the main markers of fermentative aroma. The total concentration of esters is an indicator of the fruity aroma obtained by a strain, considering that there are synergistic effects between compounds of the same chemical family [44,45]. The set of higher alcohol acetate esters represents the fruity aromas characteristic of young wines. From a sensory point of view, isoamyl acetate, which is responsible for the banana and pear flavor, is one of the most important esters.

### 3.5. PCA Applied to the Ultrasound-Treated Wines

Finally, a principal component analysis (PCA) was performed using all the polyphenolic parameters determined, and due to the high number of volatile compounds quantified, the sum of the different families of volatile compounds has been included as variables.

The objective was to find out which variables were responsible for the clustering of the wine samples after the entire winemaking process. This analysis made it possible to reduce the information provided by all the variables studied to two principal components that explain the variability of the data. The analysis showed that the wines were different. The results of the PCA can be seen in Figures 2 and 3.

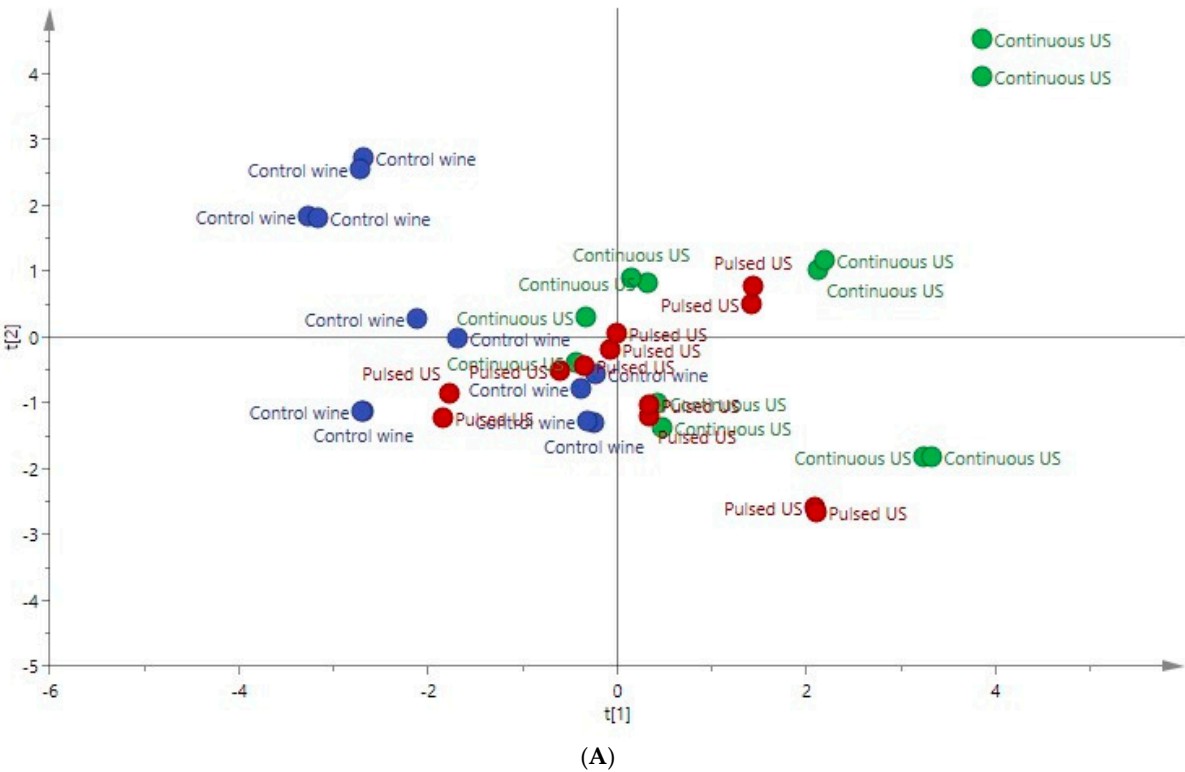

(**A**)

**Figure 2.** *Cont.*

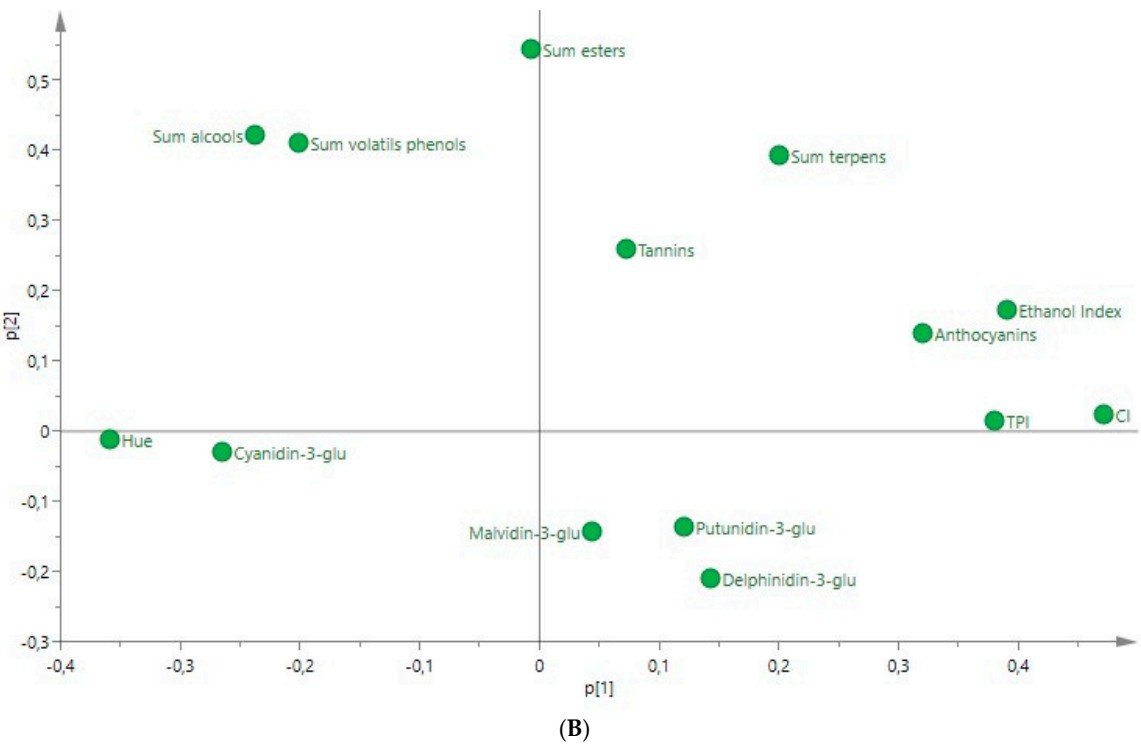

(**B**)

**Figure 2.** Rosé wine. Principal component analysis (PCA) of polyphenolic and volatile compounds in the different control and sonicated samples. (**A**) Plot of the two principal component scores. (**B**) Plot of the two principal component loadings.

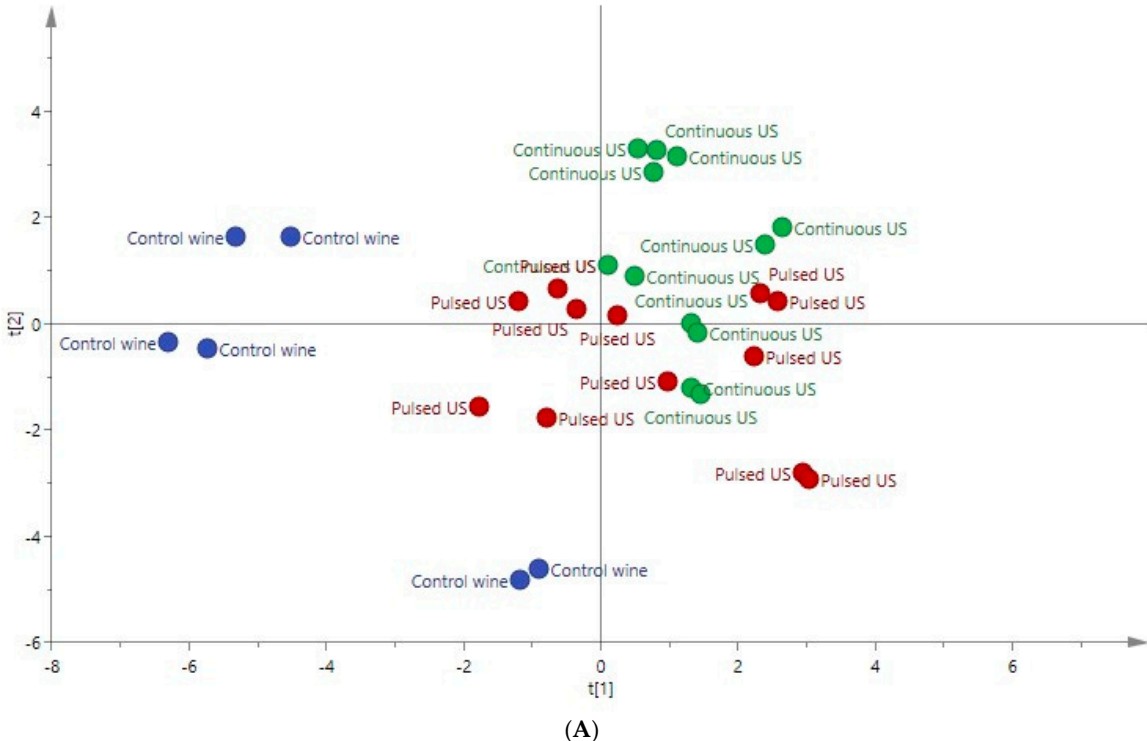

(**A**)

**Figure 3.** *Cont.*

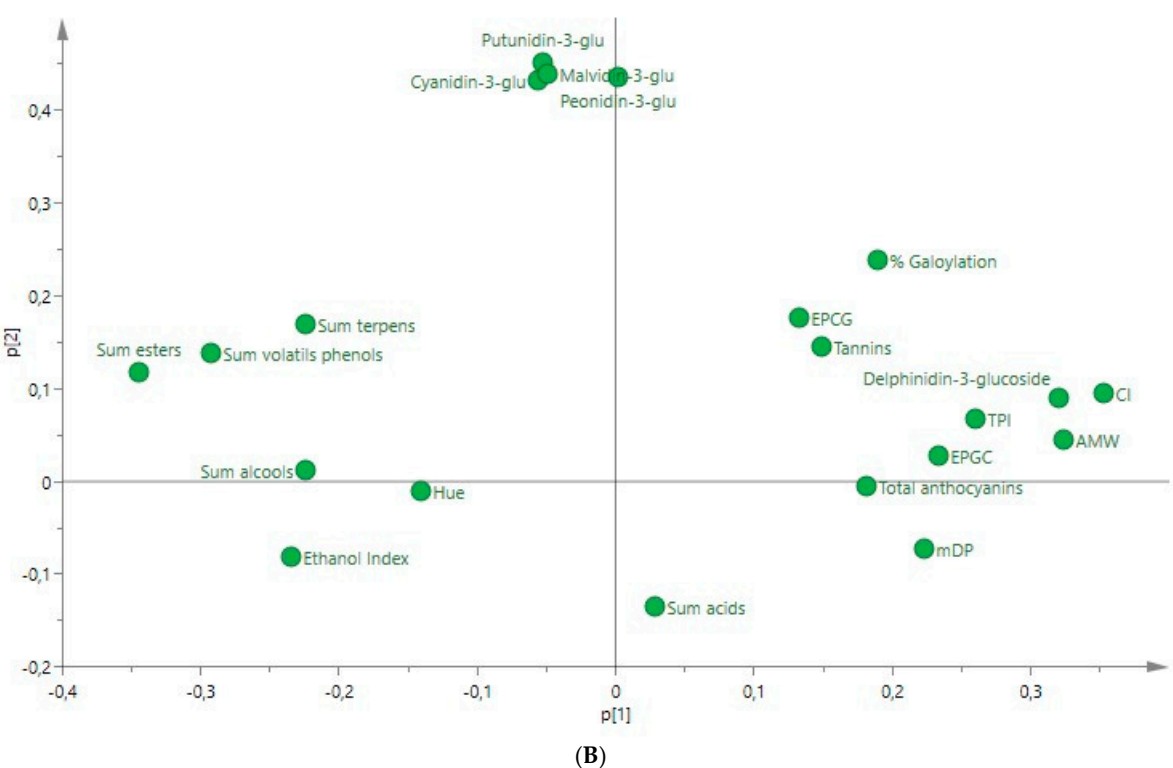

**Figure 3.** Red wine. Principal component analysis (PCA) of polyphenolic and volatile compounds in the different control and sonicated samples. (**A**) Plot of the two principal component scores. (**B**) Plot of the two principal component loadings.

In the study of rosé wines, the control wines are separated in part by PC1 (58.3%), which is related to the sum of alcohols, volatile phenols and Hue. It is not possible to separate the wines subjected to sonication from any of the PCs.

A similar behavior is obtained in the PCA for red wines (Figure 3). By means of PC1 (68.2%), most of the control wines are separated, but complete separation of the wines subjected to sonication treatment is not possible. The variables that contribute to the separation of control wines are aromatic compounds; however, sonicated wines (continuous and pulsed) are more related to polyphenolic compounds. These results are consistent with those obtained in the respective ANOVAs.

## 4. Conclusions

The results obtained in the present study show the positive effect of the application of ultrasound, both continuous and pulsed, compared to the control wine in the production of red wines, due to the increase in the polyphenolic composition. Therefore, continuous treatment in the extraction of anthocyanins and of the more condensed tannins was more effective than the pulsed treatment.

The total concentration of esters, alcohols, and volatile phenols in both rosé and red wines is lower in wines obtained with US treatment. This can be explained by the fact that the use of power or low-frequency ultrasound applied by a radiant surface from 0.400 to 25 kHz caused the spontaneous generation of heat. As for the concentration of volatile acids, it was higher in rosé and red wines sonicated with both continuous and pulsed treatment. In the case of terpenes, the effect of the US is different in the two types of wines produced. The sonicated rosé wines presented a higher concentration of terpenes, as expected; however, the opposite effect occurred in the red wines with a higher total terpene content in the controls. Future research should try different application times that are used for obtaining rose wine that are more stable over time.

**Author Contributions:** Conceptualization, V.L. and I.Á.; methodology, V.L.; software, M.J.G.-E.; validation, V.L. and M.J.G.-E.; formal analysis, I.Á.; investigation, V.L. and M.J.G.-E.; resources, V.L.; data curation, I.Á.; writing—original draft preparation, V.L. and M.J.G.-E.; writing—review and editing, V.L.; visualization, V.L.; supervision, V.L.; project administration, I.Á. All authors have read and agreed to the published version of the manuscript.

**Funding:** This research received no external funding.

**Institutional Review Board Statement:** Not applicable.

**Informed Consent Statement:** Not applicable.

**Data Availability Statement:** Data will be provided by the authors upon request.

**Conflicts of Interest:** The authors declare no conflicts of interest.

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
