# Peer review of "The Application of the Ultrasound Technique in the Production of Rosé and Red Wines"

_fermentation, doi:10.3390/fermentation10030164_

Round 1

Reviewer 1 Report

Comments and Suggestions for Authors

This is an interesting paper about the application of the ultrasound technique in the production of rosé and red wines, which has demonstrated that the aromatic composition of rose wine can be affected and that it contributes to increasing the color of red wines but without increasing the extraction of astringent tannins. Furthermore, the ultrasound treatment has favored the extraction of anthocyanins, which has had an impact on the increase of C.D. and allowed greater color stability over time. Moreover, significant differences have been found between the two US systems applied, with continuous treatment being more effective in the extraction of phenolic compounds than pulsed treatment. In my opinion, this paper might be considered for publication in the Journal of Fermentation.

The specific comments and suggestions are as follows:

1. The manuscript is relatively simple, more figures about the determination results of phenols and volatile compounds should be added in the text.

2. The ultrasound parameters should be optimized so as to get an optimal condition for the production of rosé and red wines. 

3. The mechanism of ultrasound about the effect on the aromatic composition, color etc should be deeply explored in the text.

4. Some more articles might be referred for the manuscript.

Qing-An Zhang, Hongrong Zheng, Junyan Lin, Guangmin Nie, Xuehui Fan, Juan Francisco García-Martín. The state-of-the-art research of the application of ultrasound to winemaking: A critical review[J]. Ultrasonics Sonochemistry 95 (2023) 106384

Zhen-Dan Xue, Qing-An Zhang, Ya-Feng Zhang, Er-Chun Li, Xiao Sun. Comparison of ultrasound irradiation on polymeric coloration of flavan-3-ols bridged by acetaldehyde and glyoxylic acid in model wine solution[J]. Food Chemistry, 2023(401) 134125.

Comments on the Quality of English Language

Might be modified

Reviewer 2 Report

Comments and Suggestions for Authors
